# Role of Inflammatory Factors in Mediating the Effect of Lipids on Nonalcoholic Fatty Liver Disease: A Two-Step, Multivariable Mendelian Randomization Study

**DOI:** 10.3390/nu14204434

**Published:** 2022-10-21

**Authors:** Junhong Chen, Hao Zhou, Hengwei Jin, Kai Liu

**Affiliations:** Hepatopancreatobiliary Department, General Surgery Center, First Hospital of Jilin University, Changchun 130061, China

**Keywords:** inflammatory factors, lipids, nonalcoholic fatty liver disease, Mendelian randomization, mediation

## Abstract

Aims/hypothesis: 20–80% of Nonalcoholic Fatty Liver Disease (NAFLD) have been observed to have dyslipidemia. Nevertheless, the probable mechanism of dyslipidemia’s effect on NAFLD remains unclear. Mendelian randomization (MR) was utilized to investigate the relationship between lipids, inflammatory factors, and NAFLD; and also, to determine the proportion mediated by interleukin-17(IL-17) and interleukin-1β(IL-1β) for the effect between lipids and NAFLD. Methods: Summary statistics of traits were obtained from the latest and largest genome-wide association study (GWAS). The UK Biobank provided a summary of lipid statistics, which comprised up to 500,000 participants of European descent. And NAFLD GWAS summary statistics were obtained from the FinnGen Biobank which included a total sample size of 218,792 participants of European ancestry. In order to gain an overall picture of how lipids affect NAFLD, MR with two samples was carried out. Multivariable MR determined lipids direct effects on NAFLD after adjusting for inflammatory factors, namely IL-1β, interleukin-6(IL-6), interleukin-16(IL-16), IL-17, and interleukin-18(IL-18); those lipids comprise HDL cholesterol (HDL-C), LDL cholesterol (LDL-C), triglycerides (TGs), apolipoprotein A1 (ApoA1), and apolipoprotein B (ApoB). For the purpose of determining the MR impact, an inverse variance weighted (IVW) meta-analysis of each Wald Ratio was carried out, while other methods were also performed for sensitivity analysis. Results: We discovered a positive association between genetically predicted TGs levels and a 45.5% elevated risk of NAFLD, while genetically predicted IL-1β [(IVW: OR 1.315 (1.060–1.630), *p* = 0.012) and IL-17 [(IVW: OR 1.468 (1.035–2.082), *p* = 0.032] were positively associated with 31.5% and 46.8% increased risk of NAFLD, respectively. Moreover, TG was positively associated with 10.5% increased risk of IL-1β and 17.3% increased risk of IL-17. The proportion mediated by IL-17 and IL-1β respectively and both was 2.6%, 3.1%, 14.1%. Conclusion: Genetically predicted TGs, IL-1β, and IL-17 were positively associated with increased risk of NAFLD, with evidence that IL-1β and IL-17 mediated TGs effect upon NAFLD risk. It indicated that early diet management, weight management, lipid-lowering and anti-inflammatory treatment should be carried out for patients with hyperlipidemia to prevent the NAFLD.

## 1. Introduction

Chronic liver disease is most frequently brought on by a condition known as nonalcoholic fatty liver disease (NAFLD), and majority of patients have nonprogressive nonalcoholic fatty liver [1]. About 30 and 5% of the population in the United States, respectively, are affected with nonalcoholic fatty liver disease and its subtype, nonalcoholic steatohepatitis [2,3]. It is linked to morbidity as well as mortality related to the liver, and it also raises the chance of developing cardiovascular diseases (CVDs) [4].

Multiple risk factors, including type 2 diabetes, hypertension, as well as obesity, are associated with both NAFLD and CVDs [4]. 20–80% of NAFLD-related dyslipidemia instances have been documented [5]. Nonetheless, the potential mechanism for the effect of dyslipidemia on NAFLD is still not clear. Previous studies reported that several inflammatory mediators, such as interleukin-1β (IL-1β), interleukin-6 (IL-6), and interleukin-17 (IL-17), engage in the NAFLD process of development as well as progress. Yamei Duan reported that high *C-reactive protein* (CRP), IL-1β, as well as IL-6 are significantly increase NAFLD risk [6]. Furthermore, lipid traits, such as low-density lipoprotein cholesterol (LDL-C), apolipoprotein B (ApoB), together with triglycerides (TG), are referred to as factors that increase the risk of inflammation. Based on these facts, we hypothesize that inflammatory factors mediate the effect of lipids on NAFLD. 

NAFLD is regarded to be a complicated disease feature since the disease phenotype as well as progression are both determined by interactions between the environment and a vulnerable polygenic host background. Multiple genome-wide association study (GWAS) have been carried out recently, that have enhanced our awareness of the genetics underlying NAFLD and provided the data for Mendelian randomization (MR) study. MR relies on genetic variants as exposure instrumental variables (IVs). It is employed to inquire into the link between phenotype exposure and outcome [7]. Due to randomly allocated genetic variants, MR is unlikely to have confounding biases and reverse causation inherent in observational studies [8]. These strengths also apply to mediation analysis. Throughout this work, we conducted an MR to evaluate the connection between lipids, inflammatory factors, and NAFLD; and to determine the proportion mediated by IL-17 and IL-1β for the effect between lipids and NAFLD.

## 2. Materials and Methods

### 2.1. Study Design

We employed large-scale GWAS data that was available to the public to carry out a two-sample MR analysis in order to investigate the genetic relationship between lipids as well as NAFLD. The total effect of any exposure on an outcome can be decomposed into direct and indirect effects. The direct effects of lipids on NAFLD, were obtained by multivariable MR after adjusting for inflammatory factors. The indirect effects mediated by inflammatory factors was also called Mediated effect. We carried out a two-step MR study in order to investigate whether or not inflammatory factors operate as a mediator between lipids effect on NAFLD. Firstly, we explored lipids casual effect on inflammatory factors. Next, we explored the casual effect of inflammatory factors on NAFLD. Finally, a multivariable MR was performed to determine the exposure direct effect on outcome, after adjusting for potential mediatory factors. We estimated the indirect effects (Mediated effect) by subtracting the direct effects from the total effects.

### 2.2. Data Resources for MR Analysis

Summary statistics of lipids were obtained from the (GWAS) UK Biobank [9], and included up to 500,000 participants of European ancestry. The UK Biobank has initiated a program to evaluate a vast array of biochemical indicators in baseline biological samples gathered from 2006 to 2010 from all 500,000 participants (http://www.nealelab.is/uk-biobank, accessed on 8 August 2022) [10]. 

The FinnGen Biobank provided NAFLD GWAS summary statistics, which contained a 218,792 European ancestry sample population (cases/controls: 894/217,898). FinnGen is large public-private cooperation with the objective of collecting and analyzing genetic and health data from 500,000 participants of the Finnish Biobank (https://www.finngen.fi/en, accessed on 8 August 2022).

Summary statistics for IL-1β, IL-18, IL-6, and IL-17 were acquired from GWAS of Systematic and Combined AnaLysis of Olink Proteins (SCALLOP consortium) including 21,758 participants of European ancestry [11]. Currently, the SCALLOP consortium comprises summary level data for nearly 70,000 participants from 45 cohort studies (https:olink.com/scallop/, accessed on 8 August 2022). Summary statistics for IL-16 were acquired from the GWAS of The Cardiovascular Risk in Young Finns Study (YFS) as well as FINRISK [12].

Our work did not require ethical approval because it utilized published or publicly available GWAS summary data that had already been approved by the applicable ethics and institutional review boards. Details of all GWASs included in our study are represented in Table 1.

### 2.3. Genetic IVs

Suitable single nucleotide polymorphisms (SNPs) from GWAS are substituted for characteristics in MR studies to investigate the causal connection between exposures and outcomes at the gene level. As tools, genetic variants related to statins at genome-wide significance (*p* < 5 × 10^−8^) were chosen. For all studies, the IVs were distributed independently by pruning SNPs within a 10,000 kb window and an r^2^ < 0.001 threshold [13]. Through the PhenoScanner website (http://www.phenoscanner.medschl.cam.ac.uk/ accessed on 27 August 2022) as well as the GWAS catalog (https://www.ebi.ac.uk/gwas/ accessed on 28 August 2022), we removed SNPs associated with liver inflammation, obesity, systemic infection, diabetes, and hypertension in order to limit the influence of confounding factors. Subsequently, exposure-related SNPs from the outcome datasets was obtained with minor allele frequencies > 0.01. Further, we also conducted SNP harmonization to correct the orientation of alleles. The final IVs of MR we used are presented in Appendix A. 

### 2.4. Replicative Analysis

Summary statistics of TG and NAFLD were both obtained from the (GWAS) UK Biobank as replicative analysis. Summary statistics of TG was abtained from UKB with 441,016 population of European. Number of NAFLD cases was 275 and control was 360,919. MR estimates was performed to validate the causal effect from TG on NAFLD in UKB dataset.

### 2.5. Mediation Analysis and MR Analysis

To evaluate the entire effect of lipids on NAFLD, MR was done on two samples. The overall impact of any exposure on a result can be broken down into indirect and direct effects [14]. The direct effects of lipids on NAFLD, namely HDL-C, LDL-C, TG, Apo-A1, and Apo B, were obtained by multivariable MR after adjusting for the inflammatory factors, IL-1β, IL-6, IL-16, IL-17, and IL-18. The indirect effects mediated by inflammatory factors (Mediated effect) were obtained by *β*1 × *β*2, where *β*1 are the MR effect of lipids on mediators, whereas *β*2 are the MR effect of mediators k on NAFLD adjusted for genetically determined lipids. We estimated the proportion of the mediation effect (*E*%) using the following equation [15,16]:
$$E\left(\%\right)=\frac{\sum_{k=1}^{k}\beta 1\times \beta 2k}{\sum_{k=1}^{k}\beta 3+\beta 1\times \beta 2k}$$

Here, *β*3 are the MR effect of lipids on NAFLD adjusted for genetically determined potential mediators. 

To get an MR assessment, an inverse variance weighted (IVW) meta-analysis of each Wald Ratio was conducted. These Wald Ratios were deemed most trustworthy when there was no indication of directional pleiotropy [17]. Additionally, the maximum likelihood method [18], weighted median method [19], and MR-Egger [20] were performed as complementary approaches to obtain an MR estimate. In addition, We utilized a newly developed MR approach referred to as robust adjusted profile score (RAPS), that corrects for horizontal pleiotropy using the robust adjusted contour score, hence reducing the bias caused by horizontal pleiotropy [21]. In MR-PRESSO analysis, SNPs that lead to the heterogeneity disproportionately exceeding expectations are excluded from the assessment of causal effect to reduce heterogeneity. And it was used to detect the tested and calibrated outliers [22].

### 2.6. Sensitivity Analysis

The possible pleiotropic effects of the chosen SNPs as IVs were analyzed using MR-Egger regression, and it may provide a valuable assessment of whether horizontal pleiotropy is affecting the analysis [23]. In addition, to analyze the degree of heterogeneity present among the various forms of inheritance, the Cochran Q statistic was computed, and heterogeneity was considered significant when *p* < 0.05. In addition, we performed a “leave-one-out” sensitivity assessment to find SNPs that might be significant, in which each SNP was excluded one by one from the MR. Furthermore, the forest plot, scatter plot, and funnel plot were plotted for sensitivity analysis. R version 4.0.5 along with the “Two-sample MR”, For statistical studies, the “MR-PRESSO” as well as “MR. RAPS” programs were employed. Statistics were deemed significant at *p* < 0.05.

## 3. Results

### 3.1. Selection of IVs

In the MR of lipids on NAFLD, we first obtained the following number of SNPs that that was significant in terms of genome-wide threshold (*p* < 5 × 10^−8^, r^2^ < 0.001, and kb < 10,000): LDL-C (177), HDL-C (362), ApoB (198), ApoA1 (299), and TGs (313). Two SNPs (rs58542926 and rs484032) for TGs were eliminated since they were strongly associated with diabetes. After retrieving from the outcome GWAS and harmonization, the following SNPs were used as instrumental variables: LDL-C (155), HDL-C (315), ApoB (179), Apo A1 (261), and TG (275), respectively. 

In the MR of inflammatory factors on NAFLD, we first obtained the following number of SNPs that that was significant in terms of genome-wide threshold (*p* < 5 × 10^−8^, r^2^ < 0.001, and kb < 10,000): IL-1β (23), IL-6 (23), IL-16 (3), IL-17 (13), and IL-18 (8). After retrieving from the outcome GWAS and harmonization, the following SNPs were used as instrumental variables: IL-1β (23), IL-6 (23), IL-16 (3), IL-17 (8), and IL-18 (7), respectively.

### 3.2. Total Effect of Lipids on NAFLD

A positive association between the genetically predicted TGs and an elevated risk of NAFLD of 45.5% was observed [(IVW: Odds Ratio (OR) 1.455 (1.110–1.924), *p* = 0.009]. These results were consistent with those of other methods except the MR-Egger. In addition, genetically predicted HDL-C was related with a 26.3% reduction in NAFLD risk [(IVW: OR 0.737 (0.573–0.947), *p* = 0.017]. However, according to the IVW assessment, genetically predicted LDL-C, Apo A1, as well as Apo B were not significantly linked with NAFLD risks. The results obtained using all methods are illustrated within Figure 1 and Appendix A. The MR-PRESSO method results were also consistent (Table 2). There was pleiotropy but heterogeneity was absent (Table 3). 

And We also found that the genetically predicted TGs were positively associated with increased risk of NAFLD with IVW method by replicative analysis (Appendix A). These results were consistent with those of other methods. There was no pleiotropy and heterogeneity in this study (Appendix A).

### 3.3. Causal Effect of Inflammatory Factors on NAFLD

We found that genetically predicted IL-1β [(IVW: OR 1.315 (1.060–1.630), *p* = 0.012) and IL-17 [(IVW: OR 1.468 (1.035–2.082), *p* = 0.032] were positively associated with 31.5% and 46.8% increased risk of NAFLD, respectively. In contrast, the IVW estimate showed that genetically predicted IL-6, IL-16, and IL-18 were not significantly associated with NAFLD risks. The results obtained using all methods are presented in Figure 2 and Appendix A. MR-PRESSO findings were also harmonized (Table 2). Pleiotropy and heterogeneity were absent (Table 3). 

### 3.4. Casual Effect of TGs on IL-1β and IL-17

We observed that the genetically predicted TGs were positively associated with 10.5% increased risk of IL-1β [(IVW: OR 1.105 (1.044–1.169), *p* < 0.001] and 17.3% increased risk of IL-17 [(IVW: OR 1.173 (1.034–1.331), *p* = 0.013]. The results obtained using all methods are presented in Figure 3 as well as Appendix A. MR-PRESSO findings were also consistent (Table 2).

### 3.5. Mediated Effect and Proportion by IL-1β and IL-17

In TG–IL-1β–NAFLD multivariable MR analysis, the direct effect was attenuated to OR 1.214 (95% CI: 1.012, 1.410, *p* = 0.019, Table 4). The proportion mediated by IL-1β was 3.1%. Similarly, in the multivariable MR analysis of TG–IL-17–NAFLD, the direct effect was attenuated to OR 1.250 (95% CI: 1.033–1.467, *p* = 0.013, Table 4). IL-17-mediated proportion was 2.6%. After adjusting for both, IL-1β and IL-17, the direct effect was attenuated to OR 1.179. The proportion mediated by IL-17 and IL-1β was 14.1%. 

## 4. Discussion

20–80% of NAFLD cases have been observed to have dyslipidemia, which can take the form of either hypercholesterolemia or hypertriglyceridemia, or both [24]. According to the findings of Cotrim et al., patients diagnosed with NAFLD in Brazil exhibited hyperlipidemia in 66.8% of their cases [25]. However, these findings are likely from observational studies that are related to a variety of potential confounding factors, including methods of data collection, population-specific genetics, and environmental exposures. Hence, within the context of this investigation, an MR was carried out to investigate the relationship between NAFLD, inflammatory factors, and lipids.

NAFLD may lead to nonalcoholic steatohepatitis and ultimately to cirrhosis or hepatocellular carcinoma. By using the PhenoScanner website (http://www.phenoscanner.medschl.cam.ac.uk/ accessed on 27 August 2022), we were able to remove SNPs that were associated with liver inflammation, obesity, systemic infection, diabetes, as well as hypertension. This allowed us to limit the impact that was caused by confounding factors. We also searched the SNPs we selected in the GWAS catalog (https://www.ebi.ac.uk/gwas/ accessed on 28 August 2022) to extract these who were associated with the factors you suggested. Finally, the association between lipids and NAFLD was real causal effects after excluding confounding and other factors. We also used the MR-Egger regression for evaluation of SNPs selected as IVs potential pleiotropic effects, and it may provide a valuable assessment of whether horizontal pleiotropy (liver inflammation, obesity, systemic infection and so on) is affecting the analysis. Luckily, there were no horizontal pleiotropy in all analysis.

The current study found that genetically predicted TGs, IL-1β, and IL-17 were positively related to increased NAFLD risk. The individual proportions mediated by IL-1β, and IL-17 were 3.1% and 2.6%, respectively. While proportions mediated by IL-17 and IL-1β together was 14.1%. Our MR method proved that high TGs was a risk factor for NAFLD. This is consistent with other studies which were less likely to be subject to confusion bias and reverse causality. The presence of NAFLD can be identified by the aggregation of TGs in the cytoplasm of liver cells [26]. Adipose tissue is the systemic location for energy storing as TGs, that is crucial in metabolic disorders development as insulin resistance (IR) as well as (NAFLD) [27]. The aggregation of free cholesterol inside the liver is therefore lipotoxic as well as takes place as a result of the unrestricted uptake of circulating low-density lipoprotein by the hepatic salvage receptor [28]. Fatty acids are mostly stored and transported inside cells and plasma in the form of TG molecules, which are the most common type. The hepatic fatty acid metabolism is modified by both overnutrition as well as obesity, which results in TGs aggregation within hepatocytes and the clinical condition NAFLD [29]. 

NAFLD progression process is typically explained by the traditional multiple strikes’ theory. This explanation asserts that lipid buildup causes hepatic steatosis, including inflammation, and that this in turn causes NAFLD to progress [30]. Our MR supported the multiple strikes theory. Recent research found that polymorphisms in the genes IL-1 as well as IL-17 are related to NAFLD greater severity. A latest systematic review indicated IL-1 and IL-6 have been shown to have significant correlations with NAFLD [6]. However, it was based on an observational study. Our MR found that IL-1β and IL-17 were related to NAFLD increased risk. IL-1β is cornerstone mediator of both inflammatory response, which also exacerbates damage during chronic diseases [31], and also vast array genes expression involved in secondary inflammation [32]. According to an earlier study, macrophages release IL-1β, which increases hepatocyte damage and liver fibrosis. IL-17 is the founding member of a truly innovative family of inflammatory cytokines [33]. Recently, IL-17 blocking has considered effective in multiple autoimmune diseases modulation [34]. However, further research is required to investigate the possible pathogenic mechanism of IL-17 in NAFLD.

Several aspects of our research make it superior to others: (1) statistical data summary of exposure and outcomes were from the largest and latest GWAS, and there was no overlapping sample in our study. (2) Strict criteria were established to select IVs which may strengthen the statistic power. (3) Because genetic variants were scattered over multiple chromosomes, it’s possible that potential gene-gene interactions won’t have much effect on the results. (4) In order to increase the accuracy of the estimation, this research excluded SNPs that were connected to possible confounding factors and made use of a variety of other methods for sensitivity analysis.

This study does have some restrictions: (1) The GWAS used included only the European-ancestor population. Thus, additional studies that explore the mediated effect in non-European population should be performed. (2) There were some heterogeneities in the analysis. Because GWAS data were used, it was not possible to investigate the possibility of nonlinear connections or of stratification effects that varied according to age, health status, or gender. (3) NAFLD has many subtypes, and whether the results of subtype analysis are consistent with our study needs further research. (4) The precise characterization of the causal links of the exposures a priori was essential to the success of the mediation analysis because statistical methods are unable to differentiate between the two concepts of mediation and confounding. (5) Limited lipid traits and inflammatory factors were assessed. Analysis on additional kinds should be performed to explore the other potential mediatory factors. 

## 5. Conclusions

There was evidence that IL-1β and IL-17 mediated the influence of TGs on NAFLD risk. Genetically predicted TGs, IL-1β, and IL-17 were positively related with increased risk of NAFLD. It indicated that early diet management, weight management, lipid-lowering and anti-inflammatory treatment should be carried out for patients with hyperlipidemia to prevent the NAFLD.

## Figures and Tables

**Figure 1 nutrients-14-04434-f001:**
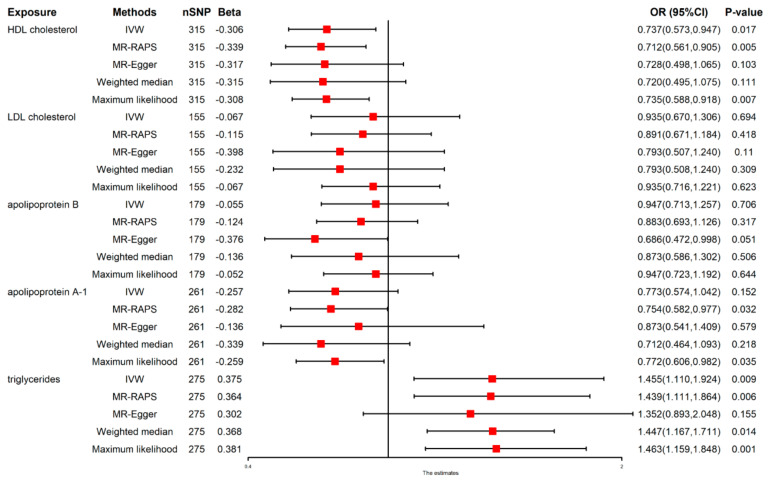
Causal effect of lipids on NAFLD. The red point means the effect (Beta).

**Figure 2 nutrients-14-04434-f002:**
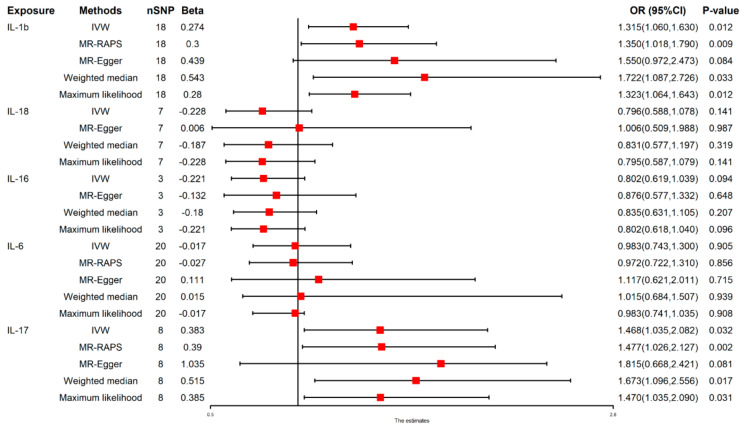
Causal effect of flammatory factor on NAFLD. The red point means the effect (Beta).

**Figure 3 nutrients-14-04434-f003:**
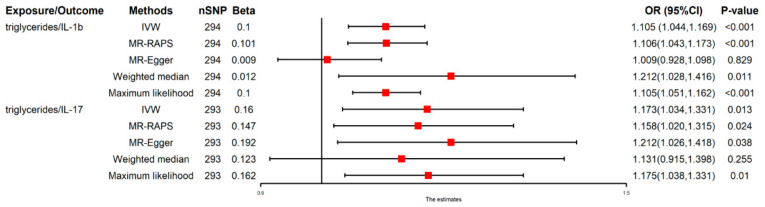
Casual effect of Tg on IL-1β and IL-17. The red point means the effect (Beta).

**Table 1 nutrients-14-04434-t001:** Details of GWAS included in Mendelian randomization analyses.

Trait	Consortium	Ethnicity	Sample Size
NAFLD (n, %)	FinnGen Biobank	European	218,792
NAFLD (n, %)	UK Biobank	European	/
HDL cholesterol (mmol/L)	UK Biobank	European	403,943
LDL cholesterol (mmol/L)	UK Biobank	European	440,546
apolipoprotein B (mmol/L)	UK Biobank	European	439,214
apolipoprotein A-1 (mmol/L)	UK Biobank	European	393,193
triglycerides (mmol/L)	UK Biobank	European	441,016
Interleukin-1β(mmol/L)	SCALLOP consortium	European	21,758
Interleukin-18 (mmol/L)	SCALLOP consortium	European	21,758
Interleukin-16 (mmol/L)	YFS/FINRISK	European	3,483
Interleukin-6 (mmol/L)	SCALLOP consortium	European	21,758
Interleukin-17 (mmol/L)	SCALLOP consortium	European	21,758

NAFLD, nonalcoholic fatty liver disease; SCALLOP, Systematic and Combined Analysis of Olink Proteins; YFS, Cardiovascular Risk in Young Finns Study.

**Table 2 nutrients-14-04434-t002:** MR-PRESSO estimates between exposures and outcomes.

Exposure Traits	Outcome Traits	Raw Estimates			Outlines Corrected Estimates		
		N	Beta	*p*-Value	N	Beta	*p*-Value
LDL-C	NAFLD	155	−0.081	0.631	151	−0.115	0.422
HDL-C	NAFLD	315	−0.318	0.016	312	−0.334	0.007
apolipoprotein B	NAFLD	179	−0.052	0.721	176	−0.088	0.502
apolipoprotein A1	NAFLD	261	−0.237	0.116	258	−0.278	0.381
triglycerides	NAFLD	275	0.32	0.022	269	0.331	0.013
IL-1β	NAFLD	18	0.273	0.016	NA	NA	NA
IL-18	NAFLD	7	−0.228	0.293	NA	NA	NA
IL-16	NAFLD	NA	NA	NA	NA	NA	NA
IL-6	NAFLD	20	−0.017	0.904	NA	NA	NA
IL-17	NAFLD	8	0.384	0.021	NA	NA	NA
triglycerides	IL-1β	294	0.106	0.0002	293	0.108	0.0001
triglycerides	IL-1β	293	0.162	0.012	291	0.166	0.007

LDL-C, LDL cholesterol; HDL-C, HDL cholesterol; IL-1β, Interleukin 1β; IL-18, Interleukin 18; IL-16, Interleukin 16; IL-6, Interleukin 6; IL-17, Interleukin 17; NAFLD, nonalcoholic fatty liver disease; NA, not available.

**Table 3 nutrients-14-04434-t003:** Heterogeneity and pleiotropy analysis.

Exposure	Outcome	Heterogeneity			Pleiotropy	
		Method	Cochran’s Q	*p*-Value	Egger-Intercept (95%CI)	*p*-Value
HDL cholesterol	NAFLD	IVW	404.387	<0.001	0.001 (−0.001,0.002)	0.937
LDL cholesterol	NAFLD	IVW	238.420	<0.001	0.016 (−0.021,0.037)	0.069
apolipoprotein B	NAFLD	IVW	261.564	<0.001	0.017 (−0.020,0.038)	0.124
apolipoprotein A1	NAFLD	IVW	399.391	<0.001	−0.005 (−0.018,0.008)	0.525
triglycerides	NAFLD	IVW	397.854	<0.001	0.003 (−0.009,0.015)	0.641
IL-1β	NAFLD	IVW	14.263	0.579	−0.030 (−0.090,0.030)	0.447
IL-18	NAFLD	IVW	1.038	0.959	−0.040 (−0.140,0.060)	0.485
IL-16	NAFLD	IVW	0.403	0.525	−0.030 (−0.130,0.070)	0.697
IL-6	NAFLD	IVW	17.814	0.468	−0.015 (−0.075,0.045)	0.632
IL-17	NAFLD	IVW	3.729	0.810	−0.081 (−0.181,0.019)	0.208
triglycerides	IL-1β	IVW	362.882	0.003	0.004 (−0.001,0.009)	0.173
triglycerides	IL-17	IVW	302.256	0.313	−0.001 (−0.005,0.003)	0.676

IL-1β, Interleukin 1β; IL-18, Interleukin 18; IL-16, Interleukin 16; IL-6, Interleukin 6; IL-17, Interleukin 17; NAFLD, nonalcoholic fatty liver disease; IVW, inverse variance weighted.

**Table 4 nutrients-14-04434-t004:** Multivariate MR analysis of the direct effect of IL-1β and IL-17 on NAFLD.

Exposure/Outcome	Adjusted Factors	Multivariate MR Analysis			Mediation Effect (%)
		nSNP	OR (95%CI)	*p*-Value	
triglycerides/NAFLD	None	275	1.455 (1.110,1.924)	0.009	
triglycerides/NAFLD	Interleukin-1β	287	1.214 (1.012,1.410)	0.019	3.1
triglycerides/NAFLD	Interleukin-17	276	1.250 (1.033,1.467)	0.013	2.6
triglycerides/NAFLD	Interleukin-1β, Interleukin-17	288	1.197 (0.794,1.600)	0.224	14.1

NAFLD, nonalcoholic fatty liver disease; OR, odds ratio; CI, confidence interval.

## Data Availability

Please see the Appendix A.

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
