# Peer review of "Role of Inflammatory Factors in Mediating the Effect of Lipids on Nonalcoholic Fatty Liver Disease: A Two-Step, Multivariable Mendelian Randomization Study"

_nutrients, 2022, doi:10.3390/nu14204434_

Round 1

Reviewer 1 Report

The authors have tried to explore the direct and indirect association of various lipids and inflammatory factors with non-alcoholic fatty liver disease using data from genome-wide association study (GWAS) and various statistical methods.  The study is well designed and for most part well written.  

The authors can improve the quality of the manuscript by addressing the following concerns of this reviewer:

1. Writing style could be substantially improved to give more clarity to distinguish between total effect, genetic effect, direct effect, indirect effect, and mediated effect.  Table 4 does some of it but is not adequate to the readers.

2. Quality of figures can be improved.  For e.g., fonts are so small that it is hard to read X-axis label in some of the figures.  Figure 3 cannot be read with ease at all.

3. The word "lipid" in the title and elsewhere may be changed to "lipids" since you are studying HDL-C, LDL-C, TGs, apolipoproteins, although your study focuses on TGs.

4. Define E(%) on line 108.  You should not give scope to readers to assume or imply.  Clarity to writing style is very important.

5. On line 115 change "To" to "to".

Author Response

Dear Trista Nie

Thank you for your revision letter, in which you encouraged us to revise our manuscript entitled “Role of Inflammatory Factors in Mediating the Effect of Lipids on Nonalcoholic Fatty Liver Disease: A Two-step, Multivariable Mendelian Randomization Study”.

We have already revised our manuscript by the suggestions from the reviewers. And we also sent it to the native English speakers to polish our manuscript this time. Please see the attached files of “Certificate OF English Editing”. And we also have reduced repetition rate. Please see the “Duplicate checking report”.

          Please find attached the revised version (nutrients-1969083-Revised manuscript) with highlighted changes of our manuscript and point by point, to the reviewers’ comments.

          We would like to thank the editors and the reviewers for their comments and recommendations that have greatly improved the quality of this paper. We hope our responses are satisfactory.

                          Sincerely,     

                         Kai Liu, M.D., PH.D 

Responds to reviewer’s comments:

Reviewer 1

Comment 1: Writing style could be substantially improved to give more clarity to distinguish between total effect, genetic effect, direct effect, indirect effect, and mediated effect.  Table 4 does some of it but is not adequate to the readers.

Response: Thanks for your suggestions. We performed a two-sample MR to assess the total effect of lipids on NAFLD. The total effect of any exposure on an outcome can be decomposed into direct and indirect effects. The direct effects of lipids on NAFLD, were obtained by multivariable MR after adjusting for inflammatory factors. The indirect effects mediated by inflammatory factors were also called Mediated effect. We have also clarified them in the method section and result section. Please see the revised edition with higthed markers.

Furthermore, we have already revised and polished our manuscript by ourselves. And we also sent it to the native English speakers to polish our manuscript this time. Please see the attached files of “Certificate OF English Editing”.

Comment 2: Quality of figures can be improved.  For e.g., fonts are so small that it is hard to read X-axis label in some of the figures.  Figure 3 cannot be read with ease at all.

Response: Thanks for your suggestions. We have provided the original figures of vector in the attached files with high resolution. And we will also provide them to the editor.

Comment 3: The word "lipid" in the title and elsewhere may be changed to "lipids" since you are studying HDL-C, LDL-C, TGs, apolipoproteins, although your study focuses on TGs.

Response: Thanks for your kindly reminding. We have changed the word “lipid” to “lipids” in the title and elsewhere.

Comment 4: Define E(%) on line 108.  You should not give scope to readers to assume or imply.  Clarity to writing style is very important.

Response: Thanks for your suggestions. We have added the definition of all the letters mentioned in the equation.

Comment 5: On line 115 change "To" to "to"

Response: Thanks for your correction. We have changed “To” to “to”.

Reviewer 2 Report

The topic is of scientific and clinical relevance, the study has clear design' strenghts and draws from important data sources. 

My main concern is the potential clinical translation of the study findings due to some underlying assumptions in the design. Authors point out that studies have shown a widely variable proportion of lipid alterations in patients with NAFLD (20-80%!). This is mainly due to the definition of NAFLD, which might include from fatty liver with no inflammation at all to advanced cirrhosis. Since the present study investigates inflammation, it'd be logic to study whether the mediatory role of inflammation distinctly impacts patients with either liver inflamation (i.e., either NAFLD with transaminase alterations that are likely attributable to steatohepatitis, or definite diagnosis of NASH) or systemic inflammation (e.g., obesity, underlying infection, etc...). Thus, I'd like the authors to better deffend what they present as a series of causal effects. 

If the focus is on predicting NAFLD development, then the analysis should be restricted to young individuals with no concomitant cardiometabolic risk factors. There is no much secret in associating TG and inflammation with NAFLD in a 60-year old male with obesity and diabetes. 

1. The abstract needs extensive rewritting. No background is provided. What does "mediatory effect of inflammatory factors" refers to? The sources are not mentioned, so the reader cannot infer whether the study is based on human or experimental data, or whether patients do have NAFLD in all cases or just a subset. Etc. 

2. Please, discuss the external validity of the study. Since GWAS data come from various sources with differing ethnic background, geographical setting, and epigenetic exposure, the impact on results should be dissected to some extent. What does European ancestry exactly mean? 

3. How was NAFLD diagnosed in patients belonging to the FinnGen Biobank? 

4. "SCALLOP consortium comprises summary level data for nearly 70,000 patients and controls from 45 cohort studies". Patients with what type of diagnosis?

5. Please, provide the rationale behind "Genetic variants associated with statins at genome-wide significance (P < 5 × 10–8) were selected as instruments."

6. Provide a brief explanation of the MR-PRESSO method in Methods. 

7. At least a summary table with the basic sociodemographic, clinical, imaging and laboratory features of patients with NAFLD vs. those without NAFLD (including the % with biopsy-confirmed NASH) should be provided. 

Author Response

Dear Trista Nie

Thank you for your revision letter, in which you encouraged us to revise our manuscript entitled “Role of Inflammatory Factors in Mediating the Effect of Lipids on Nonalcoholic Fatty Liver Disease: A Two-step, Multivariable Mendelian Randomization Study”.

We have already revised our manuscript by the suggestions from the reviewers. And we also sent it to the native English speakers to polish our manuscript this time. Please see the attached files of “Certificate OF English Editing”. And we also have reduced repetition rate. Please see the “Duplicate checking report”.

          Please find attached the revised version (nutrients-1969083-Revised manuscript) with highlighted changes of our manuscript and point by point, to the reviewers’ comments.

          We would like to thank the editors and the reviewers for their comments and recommendations that have greatly improved the quality of this paper. We hope our responses are satisfactory.

                       Sincerely,     

                         Kai Liu, M.D., Ph.D.

Responds to reviewer’s comments:

Reviewer 1

Comment 1: Writing style could be substantially improved to give more clarity to distinguish between total effect, genetic effect, direct effect, indirect effect, and mediated effect.  Table 4 does some of it but is not adequate to the readers.

Response: Thanks for your suggestions. We performed a two-sample MR to assess the total effect of lipids on NAFLD. The total effect of any exposure on an outcome can be decomposed into direct and indirect effects. The direct effects of lipids on NAFLD, were obtained by multivariable MR after adjusting for inflammatory factors. The indirect effects mediated by inflammatory factors were also called Mediated effect. We have also clarified them in the method section and result section. Please see the revised edition with higthed markers.

Furthermore, we have already revised and polished our manuscript by ourselves. And we also sent it to the native English speakers to polish our manuscript this time. Please see the attached files of “Certificate OF English Editing”.

Comment 2: Quality of figures can be improved.  For e.g., fonts are so small that it is hard to read X-axis label in some of the figures.  Figure 3 cannot be read with ease at all.

Response: Thanks for your suggestions. We have provided the original figures of vector in the attached files with high resolution. And we will also provide them to the editor.

Comment 3: The word "lipid" in the title and elsewhere may be changed to "lipids" since you are studying HDL-C, LDL-C, TGs, apolipoproteins, although your study focuses on TGs.

Response: Thanks for your kindly reminding. We have changed the word “lipid” to “lipids” in the title and elsewhere.

Comment 4: Define E(%) on line 108.  You should not give scope to readers to assume or imply.  Clarity to writing style is very important.

Response: Thanks for your suggestions. We have added the definition of all the letters mentioned in the equation.

Comment 5: On line 115 change "To" to "to"

Response: Thanks for your correction. We have changed “To” to “to”.

Reviewer 2

Comment 1 The topic is of scientific and clinical relevance, the study has clear design' strenghts and draws from important data sources.

My main concern is the potential clinical translation of the study findings due to some underlying assumptions in the design. Authors point out that studies have shown a widely variable proportion of lipid alterations in patients with NAFLD (20-80%!). This is mainly due to the definition of NAFLD, which might include from fatty liver with no inflammation at all to advanced cirrhosis. Since the present study investigates inflammation, it'd be logic to study whether the mediatory role of inflammation distinctly impacts patients with either liver inflamation (i.e., either NAFLD with transaminase alterations that are likely attributable to steatohepatitis, or definite diagnosis of NASH) or systemic inflammation (e.g., obesity, underlying infection, etc...). Thus, I'd like the authors to better deffend what they present as a series of causal effects

Response: Thanks for your time and efforts spent on our manuscript. We have made some corrections in the manuscript, please see the attached files. And here are the detailed explanation.

Mendelian randomization uses genetic variants to determine whether an observational association between a risk factor and an outcome is consistent with a causal effect. And MR relies on the natural, random assortment of genetic variants. Because these genetic variants are typically unassociated with confounders, differences in the outcome between those who carry the variant and those who do not can be attributed to the difference in the risk factors [1].

As the reviewer concerned, NAFLD is a kind of liver disease caused by the build-up of fat in the liver, therefore whether all NAFLD patients can be discussed as a whole is the main concern. We agree with the reviewer that if divided into a more detailed subset could be more precise. However, the GWAS data were provided by corresponding the consortiums as a whole and we have no authority to obtain more detailed subgrouping data. As the literature reports, the interactions between the environment and a susceptible polygenic host background determine disease phenotype and influence progression of NAFLD [2]. Recent years have witnessed multiple GWAS, which have enriched our understanding of the genetic basis of NAFLD and provided the data for MR study, which means that in genomics level, NALFD could be treated as a whole [2]. As an emerging method, MR analyses are increasingly applied to explore the causal relationship between various risk factors and NAFLD. Jiarong Xie performed a MR analysis to explore the associations between modifiable risk factors and NAFLD[3]. Yang Zhang performed a MR analysis using SNPs associated with habitual coffee intake in a published genome-wide association study (GWAS) and summary-level data of NAFLD[4]. We added these contents in the introduction and discussion according to your suggestion.

As you kindly mentioned, it'd be logical to study whether the mediatory role of inflammation distinctly impacts patients with either liver inflammation or systemic inflammation (e.g., obesity, underlying infection, etc...). NAFLD may lead to nonalcoholic steatohepatitis and ultimately to cirrhosis or hepatocellular carcinoma. We have made our efforts to improve it as you suggested. To avoid the influence of confounding factors, we excluded SNPs (Instrumental variables in MR) related to liver inflammation, obesity, systemic infection, diabetes, and hypertension using the PhenoScanner website (http://www.phenoscanner.medschl.cam.ac.uk/). We also searched the SNPs we selected in the GWAS catalog (https://www.ebi.ac.uk/gwas/) to extract those that were associated with the factors you suggested. Finally, the association between lipids and NAFLD was real causal effects after excluding confounding and other factors. We also used the MR-Egger regression[5] to evaluate the potential pleiotropic effects of the SNPs selected as Ivs, and it may provide a valuable assessment of whether horizontal pleiotropy(liver inflammation, obesity, systemic infection and so on) is affecting the analysis. Luckily, there were no horizontal pleiotropy in all analysis (Table 3). We have added another paragraph to discuss it in the discussion section according to your suggestion.

Similarly, we performed a two-sample, multivariable MR to explore the association between lipids, inflammatory factors, and NAFLD. We found Genetically predicted TGs, IL-1β, and IL-17 were positively associated with increased risk of NAFLD. It indicated that early diet management, weight management, lipid-lowering and anti-inflammatory treatment should be carried out for patients with hyperlipidemia to prevent the NAFLD. Furthermore, it was determined the proportion mediated by IL-17(3.1%) and IL-1β(2.6%) for the effect between lipids and NAFLD.  Please see the attached edition with high markers.

[1]JAMA. 2017;318(19):1925-1926. Doi:10.1001/jama.2017.17219

[2] J Hepatol. 2018 Feb;68(2):268-279. Doi: 10.1016/j.jhep.2017.09.003.

[3] Hepatology. 2022 Aug 16. Doi: 10.1002/hep.32728.

[4] Eur J Nutr. 2021 Jun;60(4):1761-1767. Doi: 10.1007/s00394-020-02369-z. 

[5] Eur J Epidemiol, 2017. 32(5): p. 377-389.

Comment 2: The abstract needs extensive rewriting No background is provided. What does “mediatory effect of inflammatory factors” refers to? The sources are not mentioned, so the reader cannot infer whether the study is based on human or experimental data, or whether patients do have NAFLD in all cases or just a subset. Etc

Response: Thanks for your suggestions. We have already re-writing the abstracts. We have added the background in the abstract: Dyslipidemia has been reported in 20%–80% of the cases associated with Nonalcoholic Fatty Liver Disease (NAFLD). However, the potential mechanism for the effect of dyslipidemia on NAFLD is still not clear. We performed Mendelian randomization (MR) to explore the association between lipids, inflammatory factors, and NAFLD, and to determine the proportion mediated by inter-leukin-17(IL-17) and interleukin-1β(IL-1β) for the effect between lipids and NAFLD. Furthermore, we also added the data sources in the method section of the abstract: Summary statistics of lipids were obtained from the UK Biobank, and included up to 500,000 participants of European ancestry. And NAFLD GWAS summary statistics were obtained from the FinnGen Biobank which included a total sample size of 218,792 participants of European ancestry.

please see the attached files.

Comment 3: Please, discuss the external validity of the study. Since GWAS data come from various sources with differing ethnic background, geographical setting, and epigenetic exposure, the impact on results should be dissected to some extent. What does European ancestry exactly mean?

Response: Thanks for your question. GWAS is known to lead to overestimation of genetic effect sizes owing to the phenomenon of the winner's curse, and this can lead to bias in MR. Dividing the dataset into two (or more) samples for estimation and testing can mitigate this problem. However, as for the GWAS data come from various sources, there was heterogeneity among our results. Due to the GWAS data, any potential nonlinear relationships or stratification effects which differs by health status, age, or gender cannot be examined. And we have discussed it in the limitation section.

As you kindly suggested, we have performed another one-sample MR analysis as Replicative analysis. Summary statistics of lipids and NAFLD were all obtained from the (GWAS) UK Biobank.

And we also found that the genetically predicted TGs were positively associated with increased risk of NAFLD with IVW method by replicative analysis. These results were consistent with those of other methods. There was no pleiotropy and heterogeneity in this study. We have addded these results in the manuscript and supplementary table 14, supplementary table 15. Please see the attached files.

Supplementary table 14. Effect of lipid metabolism traits (triglycerides) on NAFLD

Exposure

Outcome

MR

Methods

nSNP

Beta

OR (95%CI)

P-value

triglycerides

NAFLD

IVW

255

0.510

1.510(1.176,1.844)

0.003

MR-RAPS

255

0.659

1.933(1.925,1.940)

<0.001

MR-Egger

255

0.587

1.587(1.076,2.066)

0.025

Weighted median

255

0.554

1.554(1.143,1.935)

0.004

Maximum likelihood

255

0.511

1.512(1.177,1.846)

0.002

Supplementary table 15. Heterogeneity and pleiotropy analysis

Exposure

Outcome

 Heterogeneity

Pleiotropy

Method

Cochran’s Q

P-value

Egger-intercept(95%CI)

P-value

triglycerides

NAFLD

IVW

253.429

0.498

0.001(-0.001,0.003)

0.697

As we know, the population of GWAS included European, Asian, Mixed and so on. GWAS by famous consortium has larger sample size with European ancestry. Consider the population stratification, Restrict analyses to ethnically homogeneous groups. Thus, additional studies that explore the mediated effect in non-European population should be performed. We have discussed it in the limitation section.

Comment 4:  How was NAFLD diagnosed in patients belonging to the FinnGen Biobank? 

Response: Thanks for your question. All cases were unrelated patients that had undergone a liver biopsy as part of the routine diagnostic workup for presumed NAFLD having originally been identified due to abnormal biochemical tests (ALT and/or gamma-glutamyltransferase) and/or an ultrasonographically detected bright liver (ICD-10) (https://www.finngen.fi/en/access_results).

And it was also met the very strict inclusion and exclusion criteria and adjusted for covariance, such as age, sex and others.

Comment 5: "SCALLOP consortium comprises summary level data for nearly 70,000 patients and controls from 45 cohort studies". Patients with what type of diagnosis?

Response: Thanks for your question. We checked it with the GWAS and found it was a population-based study. Lasse Folkersen performed this meta analysis of GWAS published in “Nature Metabolism” and identified a catalog of genome-wide-significant genetic loci asoociated with IL-1β, IL-18, IL-6, and IL-17[1]. So we have corrected this sentence in the manuscript: “SCALLOP consortium comprises summary level data for nearly 70,000 participants”. And the detailed information of the cohort studies was prensented in the supplementary materials of this GWAS[1].

[1]Nat Metab, 2020. 2(10): p. 1135-1148.

Comment 6: Please, provide the rationale behind "Genetic variants associated with statins at genome-wide significance (P < 5 × 10–8) were selected as instruments."

Response: In the two-sample Mendelian randomization, the weak instrumental variable may lead to regression dilution bias, thus underestimating the exposure outcome relationship. Therefore, in GWAS, strict importance thresholds are often used when selecting tools. For example, the p value of 5 x 10-8 is related to the F value of about 29, which is far greater than the threshold value traditionally used to determine weak tools with F>10, meeting the validity of weak tool variables[1].

In addition, in order to illustrate the applicability of this threshold, we found that the genome-wide significance (P < 5 × 10–8) was used in the literatures published in the journals of high score influencing factors[2-5].

[1] medRxiv, 2021,book.

[2] Eur Heart J. 2021 Sep 7;42(34):3349-3357.

[3] Eur Heart J. 2020 Jan 7;41(2):221-226.

[4] Circulation. 2019 Jan 8;139(2):256-268. 

[5] Int J Cardiol. 2022 May 15;355:15-22.

[6] J Am Heart Assoc. 2021 Nov 2;10(21):e022286.

Comment 7: Provide a brief explanation of the MR-PRESSO method in Methods.

Response:Thanks for your suggestions. We have already provided a brief explanation of the MR-PRESSO method. “In MR-PRESSO analysis, SNPs that lead to the heterogeneity disproportionately exceeding expectations are excluded from the assessment of causal effect to reduce heterogeneity. And it was used to detect the tested and calibrated outliers”[1].

[1] JAMA Psychiatry, 2019. 76(4): p. 399-408.

Comment 8: At least a summary table with the basic sociodemographic, clinical, imaging and laboratory features of patients with NAFLD vs. those without NAFLD (including the % with biopsy-confirmed NASH) should be provided.

Response: The baseline data must be comparable in the GWAS. As you mentioned, it would be better to provide a summary table with the basic sociodemographic, clinical, imaging and laboratory features in our MR study. The basic information you suggested were presented in the original GWAS. Unlikely, we are not avaliable to get the related information. In order to protect the patients’ privacy, all personalized data has been encrypted. However, we noticed this GWAS was passed quality check and adjusted for the covariant according to the description of the author. Besides, to avoid bias caused by the heterogeneity of enrolled GWAS, with the reviewers’ suggestion, we also added another GWAS data to replicate our findings, and the results (seen in Response to comment 3) were consistent with our previous findings, which could also indirectly address the reviewer’s concern. And we have discussed it in the limitation section.
